# Long-Term Recreational Football Training and Health in Aging

**DOI:** 10.3390/ijerph17062087

**Published:** 2020-03-21

**Authors:** Esther Imperlini, Annamaria Mancini, Stefania Orrù, Daniela Vitucci, Valeria Di Onofrio, Francesca Gallè, Giuliana Valerio, Giuliana Salvatore, Giorgio Liguori, Pasqualina Buono, Andreina Alfieri

**Affiliations:** 1IRCCS SDN, 80143 Naples, Italy; imperlini@ceinge.unina.it (E.I.); giuliana.salvatore@uniparthenope.it (G.S.); 2Department of Movement Sciences and Wellbeing, Università Parthenope, 80133 Naples, Italy; annamaria.mancini@uniparthenope.it (A.M.); orru@uniparthenope.it (S.O.); danielavitucci@libero.it (D.V.); francesca.galle@uniparthenope.it (F.G.); giuliana.valerio@uniparthenope.it (G.V.); giorgio.liguori@uniparthenope.it (G.L.); 3CEINGE-Biotecnologie Avanzate, 80131 Naples, Italy; 4Department of Science and Technology, Università Parthenope, 80143 Naples, Italy; valeria.dionofrio@uniparthenope.it

**Keywords:** long-term recreational football training, health, aging, older, exercise, oxidative metabolism, DNA repair, senescence suppression, autophagy, VO_2_max

## Abstract

This narrative review aims to critically analyze the effects of exercise on health in aging. Here we discuss the main clinical and biomolecular modifications induced by long-term recreational football training in older subjects. In particular, the effects induced by long-term recreational football training on cardiovascular, metabolic and musculo-skeletal fitness, together with the modifications in the muscle expression of hallmarks related to oxidative metabolism, DNA repair and senescence suppression pathways and protein quality control mechanisms will be provided. All these topics will be debated also in terms of preventing non-communicable metabolic diseases, in order to achieve successful aging over time.

## 1. Introduction

Aging is characterized by a progressive, physiological decline of several biological functions leading to senescence, with a reduction of an organism’s ability to adapt to metabolic stress [1]. In fact, the organ capacity to react to perturbations and to return to homeostasis, also defined as ‘organ reserve’, decreases with age, thus explaining the structural and/or functional deterioration of different organs/systems leading to metabolic disorders, cardiovascular (CV) system impairment, muscle functional decline and osteoporosis [2]. Therefore, aging-related metabolic disorders could explain the etiology of non-communicable diseases such as type2 diabetes mellitus (T2DM), hypertension, stroke and Insulin Resistance (IR), a component of metabolic syndrome (MS) commonly observed in aging [3]. Furthermore, aging also accounts for modifications in body composition, with increased percentage of abdominal fat, associated to an increase in circulating pro-inflammatory cytokines, which in turn contribute to the onset of both IR and MS [4]. In addition, the VO_2_max, the gold international standard for estimating CV fitness, decays at about 5–10% per decade during aging; reduction in VO_2_max is closely associated with increased CV disease and mortality and to impairment in many physical abilities, including activities of daily living (ADL) [5,6,7].

Musculo-skeletal functional decline occurring in older subjects, also called *sarcopenia*, is characterized by loss of skeletal muscle mass, leading to impaired performance [8]. In particular, during the aging process, a progressive decrease (up to 30–40%) in the number of muscle fibers is observed [9]. The muscle fiber composition is also affected during aging, with a relative decrease in type2 and increase in type1 fiber numbers [10]. In addition, a remodeling of the motor units occurs during aging, resulting in reduced force-generating capacity of skeletal muscle [8,11]. Further, *sarcopenia* is associated with impairment of the mitochondrial function: reactive oxygen species (ROS), strongly increased in the muscle cells of older subjects, lead to an increased number of mutations in mitochondrial DNA, resulting in reduced oxidative capacity [12]. All these alterations are responsible for the age-related disabilities and the decrease in the quality of life (QoL) observed in older subjects [13,14].

Osteoporosis is characterized by a reduction of bone mass (i.e., Bone Mineral Density, BMD) with a bone micro-architectural deterioration, resulting in increased susceptibility to fracture. Osteoporosis is increased during aging: as a matter of fact, BMD decreases with age, especially in women, with a marked decline after the menopause, whereas the incidence of fractures increases dramatically, becoming a significant public health concern [15]. About 22 million women and 5.5 million men are estimated to have osteoporosis in the European Union and 3.5 million new fractures are declared annually, with hip fractures representing approximately 17 % of all fractures. Costs resulting from fractures or bone fragility are estimated at about EUR 37 billion and, due to the aging of the population, will dramatically increase by 25% by 2025 [16]. In addition, physical disabilities and chronic pain, the latter arising after fractures, lead to loss of independence and, eventually, to long-term care. Prevention of osteoporosis and osteoporotic fractures would have a significant positive impact on public health and on the global health economy [16].

At cellular level, some recent studies have investigated the molecular events controlling cell growth and senescence pathways, in order to better understand the mechanisms, such as the transcriptional regulation, the mitochondrial oxidative metabolism and the DNA repair pathways that could be affected during aging [17,18]. Moreover, in the aging process, the increase in protein damage, due to reduced expression of Heat Shock Proteins (HSP) and to a compromised autophagic response, with consequent deterioration of protein quality control pathways, has been demonstrated [19,20,21]. However, so far there are few data reporting the effects induced by acute and chronic exercise on the expression in skeletal muscle cells of crucial markers related to successful aging in older subjects [22,23,24,25,26]. 

Actually, the older population (adults aged 65 and over) is considerably increasing and its number is predicted to grow by more than half by 2050 [27]. Living longer implies an increase in chronic diseases as well as in age-related mental diseases [28,29]. Therefore, there is a growing need to focus on the preventive strategies aiming to assess but also improve aging status. This point of view is to be preferred over the previous standpoint based on curative approaches and on the search for disability-related risk factors [30,31]. As a matter of fact, since its first appearance, the multi-dimensional concept of successful aging, initially defined by Rowe and Kahn and referring to a complex interplay among several biopsychosocial factors in order to improve and maintain a good level of physical and mental health in older subjects, has been continuously reiterated in reports on aging of the World Health Organization (WHO) [27,30,32,33].

In this scenario, risk factors, such as sedentary lifestyle, smoking and alcohol abuse, and protective ones, like active-life and sport, play a key antagonistic role in determining quality of life (QoL) and general well-being [34]. 

## 2. Exercise and Aging 

There is a growing body of evidence that exercise contributes to preserving a good QoL across the lifespan, including cognitive performance, thus preventing morbidity and several disabilities in later life [35]. Numerous cross-sectional and longitudinal studies, in fact, have reported the association between physical fitness levels and successful aging [29,36,37,38,39,40]. In particular, higher levels of physical fitness are associated with reduced risk of obesity and type2 diabetes, CV diseases, breast and colon cancers and age-related mental diseases, such as dementia, cognitive decline, memory failure and Alzheimer’s disease [41,42,43,44]. 

Despite the strong positive association between an active lifestyle and successful aging, their specific interplay originates from an intricate balance among several factors. In fact, it is important to define the optimal volume of physical activity (time x intensity x number of sessions) for successful aging. The WHO recommends for older adults at least 150 min/week of moderate (3 < METs < 5.9) or at least 75 min/week of vigorous (METs > 6) aerobic exercise, that can be sub-divided into bouts of 10 min each, and muscle-strengthening exercises on 2 d/w or more [39]. The American College of Sports Medicine (ACSM) and the American Heart Association (AHA) encourage older adults to practice a combination of aerobic (moderate intensity, 5 d/w, 30 min/d), or alternatively vigorous intensity (3 d/w, 20 min/d), resistance (2 d/w, non-consecutive, involving the main muscle groups), flexibility (2 d/w, non-consecutive, 10 min/d) and balance training to improve health, physical fitness and QoL [40]. Recent studies have also analyzed the effects of moderate to vigorous physical activity (MVPA), measured by accelerometry, on successful aging [35,41,45,46]. They have found a positive association between the MVPA and cognitive/motor function and cardiometabolic health. Moreover, other studies have analyzed the cardiometabolic effects of intermittent MVPA exercise and its benefit on reduction of relative risk of comorbidity and mortality [47,48,49]. 

Despite the above mentioned recommendations, less than 50% of older people are able to engage in vigorous and/or long-term exercise [49]. In this context, it is important to plan a personalized Adapted Physical Activity (APA) protocol to prevent functional decline and to maintain general well-being in older adults. 

The prescription of a minimal dose of exercise is required for sedentary older adults who are not able to exercise vigorously. In this case, Hupin et al. demonstrated that 15 min/day of MVPA is a reasonable dose to gradually reach the current recommendations [50]. Nevertheless, further studies are needed to define the optimal dose of physical activity to predict successful aging over time. 

## 3. Football and Health 

Although the main guidelines for physical exercise, including those of WHO and ACSM, encourage sedentary individuals and patients to engage in exercise activities such as jogging, cycling and gym training, specific guidelines which may emphasize the health benefits of sports activities, like team sports such as football, are rather scarce [51]. 

A meta-analysis highlighted the beneficial effects of sports declaring that ‘the best evidence was found for football and running’, and that ‘evidence for health benefits of other sport disciplines was either inconclusive or tenuous’ [52]. Since then, many reports have emphasized the beneficial effects, in terms of health and well-being, of ball games in sedentary adults: in this scenario, the contribution to health benefits of recreational football training is even stronger [5,53,54,55,56,57]. 

A large body of evidence indicates recreational football as an effective tool to improve healthy status at all ages, representing a milestone for the prevention and the management of several chronic non-communicable diseases [58,59]. In particular, many studies have demonstrated that football training, like other team sports, with high intensity anaerobic actions interspersed with periods of low intensity recovery, is an intermittent exercise, effective to improve CV, musculo-skeletal and metabolic fitness, including neuromuscular and cognitive functions, in all subjects who practice it, including older subjects [5,51,53,55,58,59].

Awareness of this topic has brought about the “Football is medicine” aphorism, thus promoting interest in other team sports [51]. A recent editorial focused on the beneficial effects on fitness, health, motivation and social life provided by team sports [60]. Three studies from the University of Southern Denmark and the University of Copenhagen were examined: one related to small-sides team handball for women, one focused on basketball (both full- and half-court) and the latter focused on European football (soccer) and its possible lifelong benefits [61,62,63]. The results of these studies evidenced that: women who played handball showed an improvement in endurance and bone mineral density when compared to a sedentary control group, together with strong motivational and social scores; older players showed greater bone mineral density than non-active age-matched controls and, surprisingly, even compared to sedentary young men, demonstrating the benefits of lifelong physical activity. Thus, the conclusions of these studies suggest that in general “team sports offer a social and motivational way for improving fitness and health”. 

This statement was also confirmed by a recent pilot study conducted in order to evaluate the effects of short-term recreational team handball on physical fitness, CV and metabolism in adult men: results suggested that handball improved physical fitness and health-related metabolic parameters, contributing to the reduction of the risk of developing lifestyle diseases [64].

Therefore, recent evidence seems to indicate that recreational ball activities (in particular football and handball) should be considered as a useful tool to counteract the effects associated with lifestyle diseases [51].

Football is the most popular sport in the world: currently, more than 500 million active participants exist, 300 million of whom are football club members. Furthermore, recreational football training promotes social interactions, which have important beneficial effects on mental and social well-being, especially in older subjects. In addition, recreational football training is also associated with a limited number of injuries, differently from competitive sports; lastly, it is fun and joyful, encouraging both motivation and active participation (the drop-out rate is very low) and a greater adherence to an active lifestyle [51].

Over the last 10–15 years, the growing scientific research on recreational football largely refers to studies conducted by P. Krustrup’s group from Southern Denmark University, who have demonstrated that long-term (not less than 1 y) training (1.5–2 session/week, about 1 h/session) improves different health-related parameters such as VO_2_max, blood pressure, muscle mass, bone mass, glucose tolerance, postural balance and body composition in both healthy and unhealthy young and older subjects [5,11,51,53,54,55,56,57,58,59,60,61,62,63,64,65,66,67,68,69,70,71,72,73,74,75,76,77,78]. Here, we summarize and discuss the most recent studies of long-term recreational football training conducted on older subjects, mainly participating in small sides games (SSG) played for 1 h twice a week. The main beneficial effects of long-term recreational football training are summarized in Table 1. 

Here we report, as an example, the experimental long-term (1y) football training protocol described in Sundstrup et al. [7]. During the first three months, each training session starts with 15 min of low intensity warm-up (according to the injury prevention program 11+; F-MARC, [79]) followed by 3 × 15 min of games (4 vs 4 or 5 vs 5 players). Game periods are interspersed with 2-min rest periods. For the next 9 months, after the warm-up, the session consists of 4 × 15 min of active play interspersed with 2 min rest periods [7].

The effectiveness of recreational football training on metabolic and CV fitness is well documented: during football training, about 20% of the total training time usually includes activities performed with high intensity, greater than 90% of maximal Heart Rate (HRmax), leading to improvements in VO_2_max in adult untrained men [53]. In fact, recreational football training performed 1 h, 2.4 times a week for 12 weeks induces CV and musculo-skeletal improvements in adults and these positive effects can be maintained over time, also by reducing training frequency to about once a week [71]. 

Recreational football training is able to prevent the physiological VO_2_max reduction caused by aging and to induce a significant improvement in CV and musculo-skeletal fitness in older football players, when compared to untrained age-matched subjects (see Table 1). 12 weeks of recreational football training (3 × 40 min/week), combined with a calorie-restricted diet, improves metabolic and CV fitness both in middle-aged and in older diabetic subjects, compared to diet group [66]. These positive results are evidenced in another randomized controlled study in which middle-aged and older pre-diabetic subjects performed a 16 weeks intervention (Football training twice weekly 30- to 60- min/sessions + Dietary advice, F + D) confirming a greater improvement in metabolic and CV health profile in F + D group compared to Dietary advice group (D) [75]. 

Further evidence is reported for long-term football training: in fact, 12 month (mo) intervention (1h twice a week for the first 4 mo and 1 h three times a week for the last 8 mo) promotes a greater improvement in metabolic and CV fitness and in body composition in older football players, compared to untrained age-matched subjects [80]. Similar findings are observed in a group of lifelong football-trained veterans (>40 years of regular football training, 2.3 ± 1.1 h a week plus one match a week) compared to untrained age-matched subjects [73], suggesting a dose-response effect of the exercise. These results are confirmed by Randers et al. [81]: they demonstrated that lifelong football training induces the most positive improvements in terms of exercise capacity (VO_2_max) and muscle aerobic capacity (type I fibers enrichment and capillarization) compared to lifelong ST group.

Long-term recreational football training also induces positive effects on neuromuscular adaptations, muscle strength, postural balance and functional ability in trained vs untrained older subjects, counteracting the age-related decline in lower limb strength in older adults, thus inducing a greater autonomy in carrying out ADLs [7]. 

Football also provides a beneficial osteogenic stimulus: its rapid sprints, stops and jumps, performed in an intermittent way, and its high ground reaction forces induce high mechanical loads and are optimal for stimulating BMD, as demonstrated by both in vitro and in vivo studies [82,83]. In fact, Kish and Mezil reported increased serum levels of bone turnover markers in short-term exercised boys and young men [84,85]. Increased BMD and bone turnover serum markers were also found in sedentary older men after a recreational football training of 4 months (mo) compared to inactive age-matched subjects and these effects were more pronounced after 12 mo of intervention (see Table 1) [65]. Moreover, Hagman reported a higher BMD in femural trochanter and in the legs of lifelong older football players compared to untrained young subjects, regardless of age [63]. The osteogenic effect of football also has an instrumental influence in the prevention/management of osteoporosis in pre/post-menopausal women [86]. Increased BMD in lumbar spine and in upper femur was reported in long-term exercised premenopausal women [87]; moreover, 15 weeks of football training allowed a significant increase in plasma levels of bone turnover markers and in leg bone mass and BMD in middle-aged sedentary hypertensive women and in untrained premenopausal women [88,89]. Positive bone health effects are also reported for middle-aged/older sedentary subjects, both males and females, with prediabetes after 16 weeks of intervention [76]. Moreover, Uth et al. reported a significant correlation among the number of single accelerations/deceleration events, the bone leg mass and the volume of high intensity running in older men with prostate cancer, trained in small sided football games (*r* = 0.59–0.65) [90]. The minimum duration (single bout/training period), together with the intensity and the frequency required to induce positive bone adaptations, are still far from being clarified. 

## 4. Muscle Aging: Molecular Mechanisms

To date, few studies have focused on the effects induced by long-term football training on the expression of muscle biomarkers related to well-being and successful aging, both in adults and in older subjects. Only recently, it has been demonstrated that long-term football training induces over-expression of hallmark biomarkers involved in mitochondrial biogenesis and oxidative metabolism in muscles from long-term football trained adult subjects [91]. Subjects recruited in this study were untrained healthy males (31 ± 5.4 y) who carried out recreational football training for 1 h, 2.4 times a week for 12 weeks and 1.3 times a week for a further 52 weeks; muscle biopsies from *vastus lateralis* were obtained at baseline and post-52-weeks of intervention [71]. Alfieri et al. demonstrated, after the training period, the up-regulation of several muscle biomarkers involved in mitochondrial biogenesis and oxidative metabolism that positively correlates with the improvement in body composition, VO_2_max and metabolic profile in these subjects, thus preventing or reducing the risk of developing metabolic diseases [91]. 

More recently, mRNA and/or protein expression levels of hallmarks involved in oxidative metabolism, DNA repair promotion and senescence suppression pathways have been investigated in muscles from lifelong football-trained older subjects (veterans, VPG), (see Table 1), compared to untrained age-matched control group (CG) [92]. Lifelong football players were engaged for more than 10 y of regular football training, trained 1 session per week (1.5 ± 0.6 h/session) and played 26 ± 12 football matches (SSG) (2 × 35 min) per year, whereas control group, CG were untrained for the past 5–10 years. Both age-matched groups were homogeneous in lifestyle and habits; in addition, dietary advice was given to both groups. 

The authors have demonstrated that lifelong football training counteracts the aging mediated decline of muscle oxidative capacity, through the up-regulation in muscle cells of biomarkers involved in mitochondrial biogenesis and oxidative metabolism. Furthermore, they have demonstrated the positive association between the increased expression of muscle molecular hallmarks and systemic improvements in veteran players, VPG (see Figure 1) [93,94]. 

Lifelong football training induces the over-expression of specific key biomarkers involved in the DNA repair mechanisms and senescence suppression pathways, such as Extracellular signal-regulated protein kinases 1 and 2 (ERK1/2), a protein kinase B (AKT), the mammalian target of rapamycin (mTOR) and the Forkhead box protein M1 (FoxM1) in the muscle from VPG compared to CG (see Figure 1). In particular, increased expression of Erk1/2 proteins, that play a crucial role in the myogenic response to exercise, was found in muscle from VPG [95], along with an increased expression of AKT protein, similarly to young subjects [96]. ERk1/2 proteins are also involved in the modulation of mTOR signaling, through the phosphorylation and inactivation of tuberous sclerosis complex 2 (TSC2) [97]. In humans, mTOR responds to mechanical stimuli (i.e., muscle contraction), although its role is still rather unclear. mTOR protein expression levels were found unchanged in muscle from VPG compared to CG [98], probably due to the high interindividual variability [99]. AkT protein over-expression, through members of class O of Forkhead transcription factors (FOXO) inhibition, induces the expression of FoxM1, involved in DNA repair and suppression senescence, suggesting that lifelong football training positively affects these mechanisms in VPG muscles [92]. 

Very recently, Mancini et al. have investigated the effects of lifelong football training on the autophagic process in the skeletal muscle from VPG and CG subjects [100]. The authors have demonstrated an increased expression of messengers associated with the autolysosomal and the proteasome activity pathways in skeletal muscle from VPG compared to CG. In particular, they found an increased expression of HSPB6, HSPB1, HSP70 and HSP90 proteins that are involved in the protein quality control pathway that in turn prevents the apoptotic process. Among these proteins, HSP90 is a critical component of the lysosomal membrane type 2A protein complex (LAMP-2A), which takes part in the chaperone-mediated autophagic pathway [101] (see Figure 1). HSP induction is a crucial homeostatic adaptation that promotes healthy aging, avoiding the accumulation of damaged intracellular proteins. Furthermore, increased expression of Autophagy Factors 5 and 12 (ATG5-ATG12) complex and the anti-apoptotic B-cell lymphoma-2 protein (Bcl-2) protein, both involved in autophagy pathway, was found in muscle from VPG vs CG. In particular, the ATG5-ATG12 complex is directly involved in the assembly of the autophagosome [102,103], whereas Bcl-2 cooperates with the ATG5 diverting the muscle cells towards the autophagy when it is up-regulated, or towards apoptosis if down-regulated [104]. Lastly, the over-expression of PSMD13, a non-ATPase subunit of the 19S regulatory complex, which contributes to the assembly and/or stability of the 26S proteasome, was also found in muscle from VPG subjects [100,105], suggesting improvement of the Ubiquitin-dependent protein degradation pathway [106] and of the process finalized for the delivery of ubiquitinated proteins [107].

Molecular studies on football training and health are rather few, to the best of our knowledge those reported in this manuscript - one conducted on adults and two on older subjects [91,92,100] -, are studies in which the volunteers were recruited by the same research group, belonging to the University of Copenhagen. In these studies, recruited subjects were asked to complete a standardized diet questionnaire and 7 days recall diary (including the annotation of vitamin supplementation, common in older subjects) to analyse the food habits of the volunteers and to obtain as homogeneous a sample as possible; furthermore, all participants received similar dietary advice during the study. In addition, it should be noted that all the volunteers were active and that the molecular markers analyzed in these studies were tissue-specific-markers, i.e., skeletal muscle. The up-expression of some of these (AMPK, NAMPT, TFAM), involved in oxidative metabolism pathway and in mitochondrial biogenesis in skeletal muscle, positively correlates with the improvement in clinical systemic markers, i.e., cardiorespiratory capacity, VO_2_max, lowest blood pressure at rest, healthier metabolic profile and musculo-skeletal fitness (lower-limb muscle strength and BMD) in trained compared to active untrained subjects. One of the limits of these studies is the low number of participants and the systemic transposition of the positive results obtained in skeletal muscle. It is possible that other factors in addition to training may influence these results. However, considering the background homogeneity of the examined subjects (lifestyle, diet, citizenship), we can assume that the positive effects observed in muscle and systemic level could be ascribed to the specific football training.

## 5. Conclusions

Over the past decade, growing evidence has demonstrated the efficacy of recreational football training in the improvement of metabolic, CV and musculo-skeletal fitness, thus promoting football as a useful tool for health promotion at all ages, also in terms of CV and metabolic non-communicable disease prevention.

Most recent studies have demonstrated that long-term football training counteracts the decline in oxidative capacity in muscle mitochondria and positively affects the expression of molecular biomarkers involved in DNA repair and senescence suppression pathways, promoting successful aging.

The strength of this review is linked to a comprehensive analysis of the clinical-metabolic and molecular studies conducted on older players of long-term recreational football. Here we provide an holistic analysis of the effects mediated both at molecular (muscle) and systemic level associated with the improvement of health in long-term football older players. The molecular studies discussed in this review represent a novelty in that, even if they only refer to a limited number of subjects, no other data are present in the literature to the best of our knowledge.

The interest in these topics lies in the promising findings deriving from reported studies that could in turn stimulate future scientific research devoted to broadening and deepening insight into the systemic improvements induced by ball-games training in older subjects mediated by muscle molecular modifications. 

Finally, we hope that this review will also focus research on the minimum volume of exercise that can induce improvement at local and systemic level for successful aging.

## Figures and Tables

**Figure 1 ijerph-17-02087-f001:**
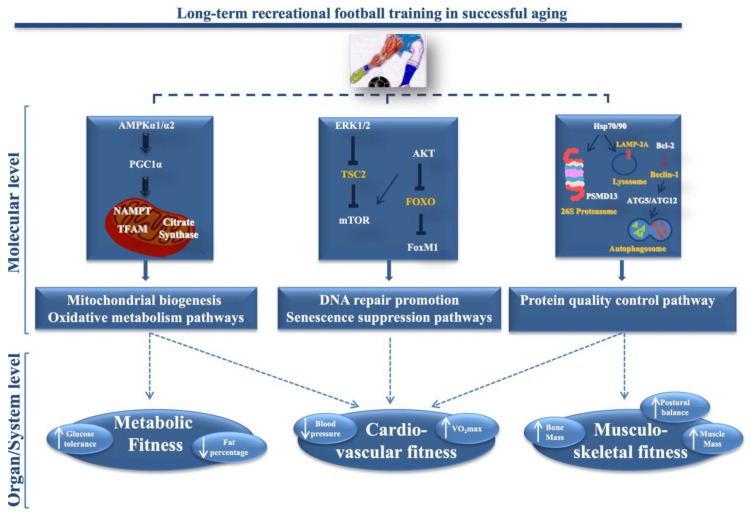
Long-term recreational football training in successful aging. Long-term recreational football training induces muscle molecular over-expression of key markers involved in the mitochondrial biogenesis and oxidative metabolism pathways (AMPKα1/α2, PGC1α, TFAM, NAMPT and citrate synthase activity), in DNA repair promotion and senescence suppression (ERK1/2, AKT, mTOR and FoxM1) and in the protein quality control pathway (HSP70/90, LAMP-2A, Bcl-2, ATG5-ATG12, PSMD13) and Systemic, Metabolic, Cardiovascular and Musculo-skeletal Fitness in long-term recreational football-trained older subjects. Abbreviations: AMPKα1/α2, Protein kinase, AMP-activated, α 1/α 2; PGC1α, Peroxisome proliferator-activated receptor-gamma coactivator-1α; TFAM, Mitochondrial transcription factor A; NAMPT, Nicotinamide phosphoribosyltransferase; ERK1/2, Extracellular signal-regulated protein kinases 1 and 2; TSC2, tuberous sclerosis complex 2; mTOR, the mammalian target of rapamycin; AKT, protein kinase B; FOXO, class O of Forkhead transcription factors; FoxM1, Forkhead box protein M1; HSP, Heat Shock Proteins; LAMP-2A, lysosomal membrane type 2A protein complex; Bcl-2, anti-apoptotic B-cell lymphoma-2 protein; ATG5-ATG12, Autophagy Factors 5 and 12; PSMD13, non-ATPase subunit of the 19S regulatory complex. Adapted from Mancini et al., 2017, 2019 [92,100] and from Krustrup & Parnell, 2019 [51].

**Table 1 ijerph-17-02087-t001:** Characteristics of selected studies on metabolic, CV and musculo-skeletal fitness in older subjects.

	Participants	Study Design	Intervention	Outcomes	Main Results
[66]	Diabetic subjects (48–68 y; *n* = 44, 17 males, 27 females)	Randomized controlled trial	Recruited subjects were randomized in: football + calorie-restricted diet group (FDG; *n* = 22) or to the calorie-restricted diet group alone (DG; *n* = 22) —12-week intervention, Small Sides Games (SSG; 3 × 40 min/week)	Body composition, VO_2_max and fasting blood samplings for metabolic parameters—lipid profile, glycemia and insulin sensitivity	Recreational football training combined with diet promote greater health benefits, with improvement in insulin sensitivity and in lipid profile in FDG compared to DG alone
[75]	Untrained subjects with prediabetes (61 ± 6y; *n* = 50, 28 males, 27 females)	Randomized controlled trial	Recruited subjects were randomized into: a Football and Dietary advice group (F + D; *n* = 27) and a Dietary advice group (D; *n* = 23)—16-week intervention; F + D performed football training (SSG, twice weekly 30- to 60- minutes each session)	Blood pressure; VO_2_max; Oral Glucose Tolerance Test (OGTT); body composition; fasting blood samplings for metabolic parameters—lipid profile and glycemia	16-week of recreational football training combined with dietary advice promotes greater improvements in metabolic and cardiovascular health profile in F + D compared to D alone
[80]	Untrained older subjects (68.2 ± 3.2 y, *n* = 26, males)	Randomized controlled trial	Recruited subjects were randomized into: Football Training Group (FTG; *n* = 9), Strength Training Group (STG; *n* = 9), or Control Group (CG; *n* = 8)—4 month (mo) and 12 mo intervention: FTG and STG performed football SSG training for 1 h/twice/week for the first 4 mo and 1 h/three times/week for the last 8 mo	VO_2_max; resting heart rate; blood pressure; cardiac function; body composition; fasting blood samplings for metabolic parameters—lipid profile and glycemia	4 mo football training elicits greater cardiovascular benefits in FTG compared to STG or CG; these positive effects are even more evident after 12 mo intervention
[73]	Veteran football Player Group (VPG; 68.1 ± 2.1 y *n* = 17 males); age-matched untrained subjects, Control Group (CG; 68.2 ± 3.2 y, *n* = 27 males).	Cross-sectional study	Recruited subjects were: lifelong football veterans, VPG: >40 years of regular football training with a total of 2.3 ± 1.1 football training session per week and a match once/week	Body composition; heart rate; blood pressure; cardiac and microvascular function, VO_2_max; fasting blood samplings for metabolic parameters—lipid profile and glycemia	Lifelong football training induces an improvement in body composition and in CV fitness in VPG compared to untrained CG subjects: VPG shows a better cardiovascular function compared with CG
[81]	Lifelong trained older subjects (65–85y; *n* = 33, males) and age-matched untrained subjects	Controlled cross-sectional study	Recruited subjects included: lifelong Soccer Players (SP, *n* = 11); Endurance-Trained (ET, *n* = 8), Strength-Trained (ST, *n* = 7) and Untrained (UT, *n* = 7) age-matched older subjects. All trained subjects were physically active 2–3 times per week during the past 40–50y	VO_2_max; resting heart rate; blood pressure; body composition; muscle fiber type and capillarization from *m. vastus lateralis* biopsies; fasting blood samplings for metabolic parameters—lipid profile and glycemia	Lifelong football training induces an improvement in exercise performance and in CV health profile in SP compared to untrained (UT) subjects; furthermore, exercise capacity and muscle aerobic capacity are even more pronounced in SP vs ST group, similarly to ET subjects
[7]	Untrained older subjects (68.2 ± 3.2 y, *n* = 27, males)	Randomized controlled trial	Recruited subjects were randomized into: recreational Football Training (FT: *n* = 10), Strength Training (ST: *n* = 9) or inactive Controls (CON: *n* = 8)—12 mo of recreational football training, SSG consisted of small-sided training sessions whereas ST consisted of high intensity exercises targeting the lower extremity and upper body	Neuromuscular adaptations; muscle strength, postural balance and functional ability	12mo (long-term) recreational football training induces positive effects on neuromuscular adaptations, muscle strength and postural balance in FT compared to control, CON, subjects. Long-term Football training also induces improvement in functional abilities, similarly to ST.
[65]	Sedentary older subjects (68.2 ± 3.2 y, *n* = 26, males)	Randomized controlled trial	Recruited subjects were randomized in: football training, F; *n* = 9; resistance training R; *n* = 9; inactive controls C; *n* = 8. 4 mo and 12 mo intervention, SSG, 2–3 times/week-45–60 min/session	Whole-body and regional (proximal femur) BMD; blood sampling for bone turnover serum markers	4 mo of recreational football training shows an osteogenic effect that is more evident after 12 mo; no effect is observed with resistance training
[63]	Lifelong football older players (65–80y, *n* = 35 males) and elite young players (18–30 y, *n* = 35, males); untrained age-matched *n* = 70, males, older and young subjects.	Controlled cross-sectional study	Recruited subjects included: lifelong football older players (FTE, *n* = 35); elite young football players (FTY, *n* = 35); untrained age-matched older (UE, *n* = 35) and young (UY, *n* = 35) subjects. FTE: >40 years of regular football training with a total of 2.3 ± 1.1 football training session per week and a match once/week; FTY: >10 years of regular football training with a total of 5.4 ± 1.6 football training session per week including matches.	Whole body and regional (femoral neck, femural trochanter and total proximal femur) BMD; blood sampling for bone turnover serum markers	Lifelong/long-term football training both in older than in elite young subjects is associated with higher BMD in the proximal femur and in whole-body with better bone turnover serum markers profile when compared to untrained age-matched subjects; furthermore, older football players show a higher BMD in femural trochanter and in legs than untrained young subjects, regardless of the age difference.
[76]	Sedentary middle-aged and older subjects with prediabetes (61 ± 9y, *n* = 50, 25 males and 25 females)	Randomized controlled trial	Recruited subjects were randomized into: a Football Training Group (FTG; *n* = 27, 14 females) and a Control group (CON; *n* = 23, 11 females)—16-week intervention; FTG performed football training twice weekly 30–60 min sessions; both groups received dietary advice.	Whole-body and regional (femur shaft and femur neck) Bone mineral content (BMC) and Bone Mineral Density (BMD); blood sampling for bone turnover serum markers	16-week of football training provides an important osteogenic stimulus improving bone health in FTG compared to untrained control subjects
[92]	Lifelong football training older subjects (Veterans, VPG, 68.2 ± 3 y, *n* = 10, males); age-matched untrained subjects (CG, *n* = 19, males).	Controlled cross-sectional study	Recruited subjects included: lifelong football players: >40 years of regular football training who trained, in the last 10y, 1 session per week (1.5 ± 0.6 h/session) and played in 26 ± 12 football matches (2 × 35 min) per year (SSG). CG participants were inactive for the past 5–10 years.	Expression of muscle molecular markers involved in oxidative metabolism/mitochondrial biogenesis (AMPKα1/α2; PGC1/α; TFAM; NAMPT), DNA repair promotion and senescence suppression pathways (ERK1/2, AKT, FoxM1) on muscle biopsies from *m. vastus lateralis.*	Lifelong football training increases the expression of key molecular muscle markers involved in oxidative metabolism, DNA repair promotion and senescence suppression pathways in VPG compared to CG subjects.
[100]	Lifelong football training veterans (65–77y, VPG, *n* = 15, males); active age-matched untrained (CG, *n* = 15, males).	Controlled cross-sectional study	Recruited subjects included: lifelong football players: >58 years of regular football training trained 1 session per week (1.5 ± 0.6 h/session) and played in 26 ± 12 football matches (2 × 35 min) per year (SSG) for the last 10 years. CG participants were inactive for the past 5–10 years.	Expression of muscle molecular markers involved in protein quality control pathways (HSP70/90; Bcl-2; PSMD13; ATG5–ATG12) on muscle biopsies from *m. vastus lateralis*.	Lifelong football training positively affects protein quality control processes in VPG compared to CG subjects.

**Abbreviations:** AMPKα1/α2, Protein kinase, AMP-activated, α 1/α 2; PGC1α, Peroxisome proliferator-activated receptor-gamma coactivator-1α; TFAM, Mitochondrial transcription factor A; NAMPT, Nicotinamide phosphoribosyltransferase; ERK1/2, Extracellular signal-regulated protein kinases 1 and 2; AKT, protein kinase B; FoxM1, Forkhead box protein M1; HSP, Heat Shock Proteins; Bcl-2, anti-apoptotic B-cell lymphoma-2 protein; ATG5-ATG12, Autophagy Factors 5 and 12; PSMD13, non-ATPase subunit of the 19S regulatory complex.

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
