# Peer review of "Long-Term Recreational Football Training and Health in Aging"

_ijerph, 2020, doi:10.3390/ijerph17062087_

Round 1

Reviewer 1 Report

Dear Editor,

thank you for the opportunity to review this manuscript, which considers some interesting, applied issues. This work gives a significant contribution to the importance of sport and physical exercise for healthy aging.

The paper is well written and comprehensively described, english is used correct and readable.

Anyway, I purpose to the authors to improve related reference, with more recent studies.

For the relation between Physical exercise and  dementia:

Iuliano, E. (2019). Physical exercise for prevention of dementia (EPD) study: background, design and methods. BMC public health, 19(1), 659

For the importance of Physical exercise to prevent cognitive decline:

di Cagno, A. (2015). Effects of different types of physical activity on the cognitive functions and attention in older people: A randomized controlled study. Experimental gerontology, 70, 105-110.

Fiorilli,G.  (2017). Twelve-week exercise influences memory complaint but not memory performance in older adults: A randomized controlled study. Journal of aging and physical activity, 25(4), 612-620.

Author Response

Point to by point reply

MS n. ijerph-695587 by Imperlini et al.

Comment Reviewer 1

General comment: This work gives a significant contribution to the importance of sport and physical exercise for healthy aging. The paper is well written and comprehensively described, english is used correct and readable.

Anyway, I purpose to the authors to improve related reference, with more recent studies. For the relation between Physical exercise and dementia:

Iuliano, E. (2019). Physical exercise for prevention of dementia (EPD) study: background, design and methods , (1), 659

For the importance of Physical exercise to prevent cognitive decline:

di Cagno, A. (2015). Effects of different types of physical activity on the cognitive functions and attention in older people: Arandomized controlled study. , , 105-110.

Fiorilli,G. (2017). Twelve-week exercise influences memory complaint but not memory performance in older adults: A randomized controlled study

We thank the reviewer for his/her opinions and suggestions.

  • According to the reviewer’s suggestions we add more updated references in the suggested topics in the section 2. Exercise and aging, line 107 (see reff n. 42,43,44 in the References).

Reviewer 2 Report

This review article narratively reviewed the evidence on football training and cardiorespiratory, muscular, bone health and their potential biomolecular mechanisms. The review brings interesting knowledge into light with a particular focus on football training. However, the information presented here are not very well organised, the literature was not systematically reviewed, there is a risk of bias in reporting. The following are my review comments.

Major comments

A systematic literature search on lifelong football training and healthy ageing should be conducted. Studies reporting effects of lifelong football training on various health outcomes (cardiorespiratory, muscular, bone health outcomes and biomolecular markers) should be summarised in a table. This should contain study design (intervention, cross-sectional), participant characteristics (N, age, gender, previous sport experience, etc), intervention characteristics (time, frequency, duration, football training program, instructor’s experience and involvement), health outcome and major findings.

Title

‘Lifelong’ indicates lifelong participation to football for at least decades. However, some review findings are based on intervention durations of one year or less. Please reconsider. ‘Healthy Ageing’ is probably too broad for this review. Please choose more appropriate summary that encompass the review’s focus.

Affiliations

The affiliations should be given in English.

Introduction

The introduction on healthy ageing should include background on cardiorespiratory, muscular and bone health.

Football and Health

Line 93. What are the other sports activities and team sports that have the possible health benefits? Describe what characteristics of football makes different among those sports activities and team sports. Lines 102-107. Please quantify the exercise intensity and amount, the balance between sprinting and resting in the referred football studies. 2 hours of football can be very different in intensity depending on the emphasis on training skills, endurance or competitiveness. Lines 121-132. This section on sarcopenia or muscular health includes no description of football studies. These are just background and should be moved to introduction. Lines 133-149. Again, these are just background and should be moved to introduction. The section should describe the relationship between football training and bone health.

Muscle Ageing: Molecular Mechanisms

This section is focused on described in the molecular biomarkers found to be associated with football training. Descriptions of study design, participant and intervention characteristics are completely lacking. These molecular biomarkers are influenced by many lifestyle factors and football training is not the only one factor. The integrity of the findings including risk of bias should be critically reviewed. The abbreviations should be spelled out in their first appearance. Line 166. Line 170. Alpha is missing. Line 167. Reference should be cited. Lines 189-191. Ref 85 is not Mancini et al. Figure 1 is fragmental. Kicking a ball is not likely the cause of these effects. The figure should describe what physiological and physical factors in football training is likely causing these effects. The link between molecular level findings and healthy ageing should be supplemented by cardiorespiratory, muscular and bone health findings.

Conclusion

Lines 217-220. This section has no original information and should be deleted.

Other

The term ‘elderly’ should be avoided. Avers D et al. Use of the term “Elderly.” J Geriatr Phys Ther 34: 153-154, 2011. The authors should clarify the term ‘lifelong’. What are the average starting age and duration to be called lifelong training? Risk of injury and negative sides of football compared to other sports should also be mentioned. The review should also highlight what is yet to be clarified in future studies. The authors claim that all of these findings are attributed to football training but it is only one of many factors. The limitations of individual studies reviewed and limitations of this review should be discussed.

Author Response

Point to by point reply

MS n. ijerph-695587 by Imperlini et al.

Comment Reviewer 2

General comments: This review article narratively reviewed the evidence on football training and cardiorespiratory, muscular, bone health and their potential biomolecular mechanisms. The review brings interesting knowledge into light with a particular focus on football training. However, the information presented here are not very well organised, the literature was not systematically reviewed, there is a risk of bias in reporting. The following are my review comments.

Major comments

-A systematic literature search on lifelong football training and healthy ageing should be conducted. Studies reporting effects of lifelong football training on various health outcomes (cardiorespiratory, muscular, bone health outcomes and biomolecular markers) should be summarised in a table. This should contain study design (intervention, cross-sectional), participant characteristics (N, age, gender, previous sport experience, etc), intervention characteristics (time, frequency, duration, football training program, instructor’s experience and involvement), health outcome and major findings.

  • We agree with the reviewer’s observations and we modify accordingly the manuscript. The analyzed studies, the recruited subjects, the study design, the intervention protocols, the outcomes and the main results are now summarized in a descriptive table (see Table 1).

---Title

‘Lifelong’ indicates lifelong participation to football for at least decades. However, some review findings are based on intervention durations of one year or less. Please reconsider.

  • We agree with the reviewer that the term” lifelong” refers to training > 10 years, whereas in our narrative review we have also considered studies with shorter intervention protocols. So, we replace the term ”lifelong” with “long-term” in the title.
  • We also add in the title the term “recreational” since the reported studies mainly focused on this type of training.

‘Healthy Ageing’ is probably too broad for this review. Please choose more appropriate summary that encompass the review’s focus.

  • According with reviewer’s suggestions, we re-write the title as follow:

“Long-term Recreational Football Training and Health in Aging”

---Affiliations

The affiliations should be given in English.

  • According with reviewer’s suggestions, we re-write the affiliation in English; only CEINGE remains in Italian as mentioned on the international web page: www.ceinge.unina.it)

---Introduction

The introduction on healthy ageing should include background on cardiorespiratory, muscular and bone health.

- We re-write part of the Introduction section, including a short background on main physiological modifications that occurr in aging. Introduction section lines: 36-74, underlined.

---Football and Health

Line 93. What are the other sports activities and team sports that have the possible health benefits? Describe what characteristics of football makes different among those sports activities and team sports.

  • We specify the characteristics of football compared to other team sports, see section Football and Health, lines: 141-160, underlined.

Lines 102-107. Please quantify the exercise intensity and amount, the balance between sprinting and resting in the referred football studies. 2 hours of football can be very different in intensity depending on the emphasis on training skills, endurance or competitiveness?.

  • According to the reviewer’s comments, we provide missing informations in several part of the “Football and Health “ section. In particular, they are reported in lines 168-177 .

Lines 121-132. This section on sarcopenia or muscular health includes no description of football studies. These are just background and should be moved to introduction.

  • According to the reviewer’s suggestions, we moved this section to the introduction section, lines 49-60.

Lines 133-149. Again, these are just background and should be moved to introduction. The

section should describe the relationship between football training and bone health.

  • According to the reviewer’s suggestions, we moved this section to the introduction section, see lines 61-74.

---Muscle Ageing: Molecular Mechanisms

This section is focused on described in the molecular biomarkers found to be associated with football training. Descriptions of study design, participant and intervention characteristics are completely lacking. These molecular biomarkers are influenced by many lifestyle factors and football training is not the only one factor. The integrity of the findings including risk of bias should be critically reviewed.

The abbreviations should be spelled out in their first appearance.

-          According to the reviewer’s suggestions, all the abbreviations are now spelled out and also appear in the Figure 1 Legend.

Line 166. Line 170. Alpha is missing.

  • Now, “α” is present in the text of Figure 1

Line 167. Reference should be cited.

  • We report the reference, as requested, in section 3.Muscle Aging: Molecular Mechanisms, line 242.

Lines 189-191. Ref 85 is not Mancini et al.

            - Unfortunately, during the creation of the PDF there has been a movement of numbers in the References that generated confusion. Now we checked carefully, after the PDF creation.

Figure 1 is fragmental. Kicking a ball is not likely the cause of these effects. The figure should describe what physiological and physical factors in football training is likely causing these effects. The link between molecular level findings and healthy ageing should be supplemented by cardiorespiratory, muscular and bone health findings.

            - According to reviewer’suggestions, we prepare a new Figure 1 in which we report the link between long-term recreational football-mediated muscle molecular effects and systemic improvements on metabolic, cardiovascular and musculo-scheletal fitness in older subjects.

---Conclusion

Lines 217-220. This section has no original information and should be deleted.

-          According to reviewer’s comments, we deleted this sentence.

---Other

The term ‘elderly’ should be avoided. Avers D et al. Use of the term “Elderly.” J Geriatr Phys Ther 34: 153-154, 2011

-          According to reviewer’s comments, we replace the term “elderly” with “older subjects” alongsite the manuscript.

The authors should clarify the term ‘lifelong’. What are the average starting age and duration to be called lifelong training?

-          All these informations are now reported in Table 1 and in the manuscript.

-Risk of injury and negative sides of football compared to other sports should also be mentioned. The review should also highlight what is yet to be clarified in future studies.

The authors claim that all of these findings are attributed to football training but it is only one of many factors. The limitations of individual studies reviewed and limitations of this review should be discussed.

            - According to reviewer’s comments, we re-write the conclusions, see Conclusion section lines: 327-347, underlined.

Reviewer 3 Report

This was a very good paper to read for me because it reminded me of things relating to aging and biological functions I do not often study. Essentially this is a literature review demonstrating the differences in biomedical research regarding footballers. This should be more clear in the abstract. Further, the authors should keep in mind that the work is complete; and therefore, is written in present tense. If the authors choose to use past tense at the beginning of a paragraph, the paragraph needs to remain in past tense (and so forth).

The grammar was the biggest issue I had throughout the manuscript but they were not issues not easily resolved. I would double check the number references associated with what is stated within the manuscript (and note one in particular in question- but I could be wrong).

The conclusion needs more information. Using one word sentences without providing additional information nor relating it back to the purpose of the review does not make for a strong conclusion.  Additionally, where is the gap in the research? With all of these studies you have read and vetted, what is missing? Where can the research go from here?

Thank you for a fun read!

Author Response

Point to by point reply

MS n. ijerph-695587 by Imperlini et al.

Comment Reviewer 3

We thank the reviewer for his/her opinions and suggestions.

General comment: Essentially this is a literature review demonstrating the differences in biomedical research regarding footballers. This should be more clear in the abstract.

According to reviewer’s comments, we re-write part of the abstract, highlighting the biomedical, metabolic and cardiovascular effects mediated by long-term football training in older subjects.

Further, the authors should keep in mind that the work is complete; and therefore, is written in present tense. If the authors choose to use past tense at the beginning of a paragraph, the paragraph needs to remain in past tense (and so forth). The grammar was the biggest issue I had throughout the manuscript but they were not issues not easily resolved. I would double check the number references associated with what is stated within the manuscript (and note one in particular in question- but I could be wrong).

-          An English spelling check throughout the manuscript was performed, especially in the use of the present and the past tense.

-          - Unfortunately, during the creation of the PDF there has been a movement of numbers in the References that generated confusion. Now we checked carefully, after the PDF            creation.

The conclusion needs more information. Using one word, sentences without providing additional information nor relating it back to the purpose of the review does not make for a strong conclusion. Additionally, where is the gap in the research? With all of these studies you have read and vetted, what is missing? Where can the research go from here?

            - According to reviewer’s comments, we re-write the conclusions, see Conclusion section   lines: 327-347, underlined.

Round 2

Reviewer 2 Report

The authors have made modifications to the manuscript in response to some of the review commends. However, some important commends have been left unanswered.

  1. A systematic literature search on lifelong football training and healthy ageing should be conducted. 
  2. "Universitia" is not English.
  3. Line 93. What are the other sports activities and team sports that have the possible health benefits? Describe what characteristics of football makes different among those sports activities and team sports.
  • We specify the characteristics of football compared to other team sports, see section Football and Health, lines: 141-160, underlined.

>There is no comparison with other sports. 

4. These molecular biomarkers are influenced by many lifestyle factors and football training is not the only one factor. The integrity of the findings including risk of bias should be critically reviewed.

5. The authors claim that all of these findings are attributed to football training but it is only one of many factors. The limitations of individual studies reviewed and limitations of this review should be discussed.

Author Response

Point-by point reply

MS n. ijerph-695587 by Imperlini et al.

 Reviewers’ comments:

It is felt that the comparison with other sports needs to be more explicit as it is currently unclear what other sports have been compared (lines 141-160). What are the other sports activities and team sports that have the possible health benefits? Describe what characteristics of football makes different among those sports activities and team sports.

Thank you for your observation. To address this issue, we have added the following text to the Football and Health section, page 3, lines: 137-155, highlighted in green: The knowledge on this topic encouraged the "Football is medicine" aphorism, thus suggesting interest in other team sports [51]. A recent editorial aimed to investigate the beneficial effects on fitness, health, motivation and social provided by team sports [62]. Three studies from the University of Southern Denmark and the University of Copenhagen were examined: one related to the small- sides team handball for women, one focused on basketball (both full- and half-court) and the latter focused on European football (soccer) and its possible lifelong benefits [63-65]. The results of these studies evidenced that: - women who played handball had an improvement in endurance and bone mineral density when compared to a sedentary control group, together with strong motivational and social scores; -older players showed greater bone mineral density than non-active age-matched controls, and surprisingly, even compared to sedentary young men, demonstrating the benefits of lifelong physical activity. Thus, the conclusions of these studies suggest that in general “team sports offer a social and motivational way for improving fitness and health”.

This statement was also confirmed by a recent pilot study conducted in order to evaluate the effects of short-term recreational team handball on physical fitness, CV and metabolism in adult men: results suggested that handball improved physical fitness and health-related metabolic parameters, contributing to the reduction of the risk of developing lifestyle diseases [66].

Therefore, recent evidence seems to indicate that recreational ball activities (in particular football and handball) should be considered as an useful tool to counteract the effects associated to lifestyle diseases [51].

New references were added to the manuscript (see reff. 62,63,64 and 66).

There should be a fuller discussion of the molecular biomarkers as they are influenced by many lifestyle factors and football training is not the only one factor. The integrity of the findings including risk of bias should be critically reviewed.

Thank you for your observation. To address this issue, we have added the following text to the Muscle Aging: Molecular Mechanisms, lines: 291-309, highlighted in green: The molecular studies on football training and health are rather few, to the best of our knowledge those reported in this manuscript - one conducted on adults and two on older subjects [94,95,103] -, are studies in which the volunteers were recruited by the same research group, belonging to the University of Copenhagen. In these studies, recruited subjects were asked to complete a standardized diet questionnaire - 7 days recall diary (including the annotation of vitamin supplementation, common in older subjects) to analyse the food habits of the volunteers and to obtain a sample as homogeneous as possible; furthermore, all participants received similar dietary advices during the study. In addition, it should be noted that all the volunteers were active and that the molecular markers analyzed in these studies were tissue specific-markers, i.e. skeletal muscle. The up-expression of some of them (AMPK, NAMPT, TFAM), involved in oxidative metabolism pathway and in mitochondrial biogenesis in skeletal muscle, positively correlates with the improvement in clinical systemic markers, i.e. cardiorespiratory capacity, VO2max, lowest blood pressure at the rest, healthier metabolic profile and musculo-skeletal fitness (lower-limb muscle strength and BMD) in trained compared to active untrained subjects. One of the limits of these studies is the handful of participants and the systemic transposition of the positive results obtained in skeletal muscle. It is possible that other factors in addition to training may influence these results. However, considering the background homogeneity of the examined subjects (lifestyle, diet, citizenship), we can assume that the positive effects observed in muscle and systemic level could be ascribed to the specific football training.
